# Hydrogen production from the air

Jining Guo[1], Yuecheng Zhang[1], Ali Zavabeti [1], Kaifei Chen[1], Yalou Guo[1], Guoping Hu [1,2] ✉, Xiaolei Fan [3,4] ✉ & Gang Kevin Li [1] ✉

Green hydrogen produced by water splitting using renewable energy is the most promising energy carrier of the low-carbon economy. However, the geographic mismatch between renewables distribution and freshwater availability poses a significant challenge to its production. Here, we demonstrate a method of direct hydrogen production from the air, namely, in situ capture of freshwater from the atmosphere using hygroscopic electrolyte and electrolysis powered by solar or wind with a current density up to 574 mA cm$^{-2}$. A prototype of such has been established and operated for 12 consecutive days with a stable performance at a Faradaic efficiency around 95%. This so-called direct air electrolysis (DAE) module can work under a bone-dry environment with a relative humidity of 4%, overcoming water supply issues and producing green hydrogen sustainably with minimal impact to the environment. The DAE modules can be easily scaled to provide hydrogen to remote, (semi-) arid, and scattered areas.

Hydrogen is the ultimate clean energy. Despite being the most abundant element in the universe, hydrogen exists on the earth mainly in compounds like water. $H_2$ produced by water electrolysis using renewable energy, namely, the green hydrogen, represents the most promising energy carrier of the low-carbon economy[1–3]. $H_2$ can also be used as a medium of energy storage for intermittent energies such as solar, wind, and tidal[4–6].

The deployment of water electrolyzer is geographically constrained by the availability of freshwater, which, however, can be a scarce commodity. More than one-third of the earth's land surface is arid or semi-arid, supporting 20% of the world's population, where freshwater is extremely difficult to access for daily life, let alone electrolysis[7,8]. In the meanwhile, water scarcity has been exacerbated by pollution, industrial consumption, and global warming. Desalination may be used to facilitate water electrolysis in coastal areas, however, substantially increasing the cost and complexity of hydrogen production. On the other hand, areas rich in renewable energies are commonly short in water supply[9]. Figure 1a and 1b shows a distinctive geographic match between the shortage of freshwater and the potential of solar power and wind power, respectively, in the majority of the continents, such as North Africa, West, and Central Asia, Midwest Oceania, and southwest of North America.

Few studies have been trying to mitigate the water shortage for electrolysis. Direct saline splitting can produce hydrogen, which, however, faces a serious challenge of handling chlorine byproduct[10,11]. Some proton/anion exchange membrane electrolyzers can use high humidity vapor feed to the anode; however, the cathode of all of these electrolyzers must operate in an air-free atmosphere[12–20], purged by an inert carrier gas such as nitrogen or argon, resulting in particularly low $H_2$ product purity of less than 2%. On another note, photocatalytic water splitting has a potential to use vapor feed[21], but the biggest problem of this method is its low solar-to-hydrogen efficiency (around 1%) in real-world demonstrations[22,23] and to make it more complicated, the product is a mixture of $H_2$ and $O_2$ gases which require an extra separation process.

In this work, we corroborate that moisture in the air can directly be used for hydrogen production via electrolysis, owing to its universal availability and natural inexhaustibility[24–28]—there are 12.9 trillion tons of water in air at any moment which is in a dynamic equilibrium with the aqua-sphere[29]. For example, even in the Sahel desert, the average relative humidity (R.H.) is about 20%[19], and the average daytime R.H. at Uluru (Ayers Rock) in the central desert of Australia is 21%[30]. Considering deliquescent materials such as potassium hydroxide, sulfuric acid, propylene glycol[31,32] can absorb water vapor from a bone-dry air,

[1]Department of Chemical Engineering, The University of Melbourne, Parkville, Vic 3010, Australia. [2]Ganjiang Innovation Academy, Chinese Academy of Sciences, Ganzhou, Jiangxi 341000, China. [3]Department of Chemical Engineering, School of Engineering, The University of Manchester, Manchester M13 9PL, UK. [4]Nottingham Ningbo China Beacons of Excellence Research and Innovation Institute, 211 Xingguang Road, 315191 Ningbo, China. ✉e-mail: gphu@gia.cas.cn; xiaolei.fan@manchester.ac.uk; li.g@unimelb.edu.au

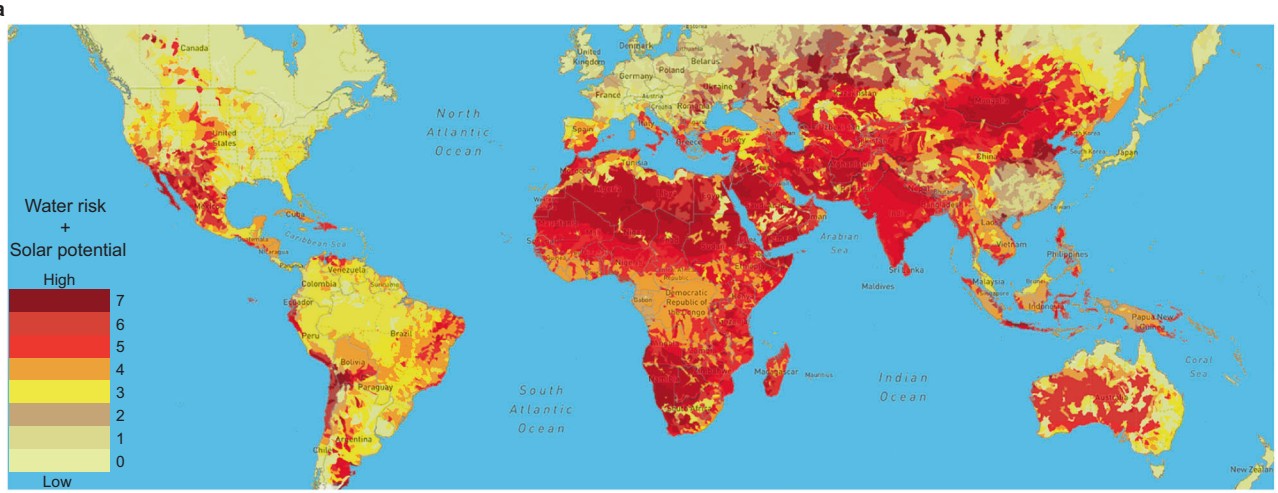

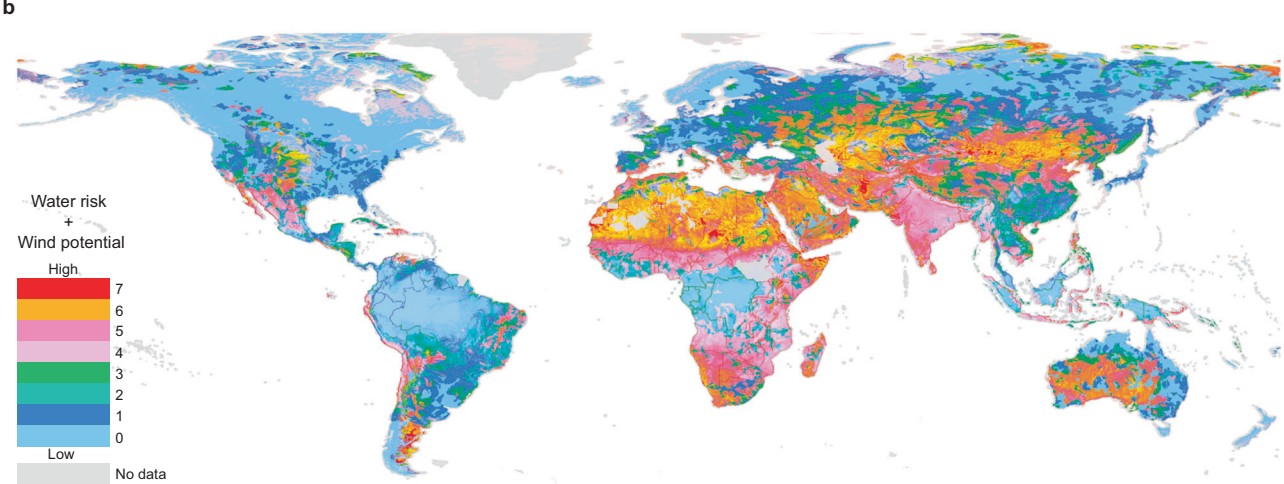

**Fig. 1 | Superimposed atlas of water risk and the renewable energy. a** water risk and solar energy potential; **b** water risk and wind energy potential (excluding coast areas). Separate maps are available in Supplementary Fig. 1–3. Source: World Resources Institute (WRI) Aqueduct[42], accessed on [04.2021], aqueduct.wri.org; World Bank Group[43], accessed on [04.2021], https://globalsolaratlas.info; Technical University of Denmark[44], accessed on [04.2021], https://globalwindatlas.info− Creative Commons Attribution International 4.0 License.

here, we demonstrate a method to produce high purity hydrogen by electrolyzing in situ hygroscopic electrolyte exposed to air. The electrolyzer operates steadily under a wide range of R.H., as low as 4%, while producing high purity hydrogen with a Faradaic efficiency around 95% for more than 12 consecutive days, without any input of liquid water. A solar-driven prototype with five parallel electrolyzers has been devised to work in the open air, achieving an average hydrogen generation rate of 745 L $H_2$ day$^{-1}$ m$^{-2}$ cathode; and a wind-driven prototype has also been demonstrated for $H_2$ production from the air. This work opens up a sustainable pathway to produce green hydrogen without consuming liquid water.

## Results

### Design of the Direct Air Electrolysis (DAE) module for hydrogen production

Hydrogen production from the air was realized through our DAE module. As shown in the sandwich structure in Fig.2a, b, this module consists of a water harvesting unit in the middle and electrodes on both sides paired with gas collectors. The module is integrated with a power supply, for example, a solar panel, a wind turbine, and any other renewable generators. Importantly, the water harvesting unit also serves as the reservoir to hold the electrolyte. Porous medium such as melamine sponge, sintered glass foam is soaked with deliquescent ionic substance to absorb moisture from the air via the exposed surfaces. The captured water in the liquid phase is transferred to the surfaces of the electrodes via diffusion and subsequently split into hydrogen and oxygen in situ which are collected separately as a pure gas, since both electrodes are isolated from air (Supplementary Figs. 4–6). The reservoir between the endplate and the porous foam (Supplementary Fig. 5b) works as an air barrier and a buffer for the volume of the ionic solution at excessive fluctuation of the air humidity. This reservoir avoids the overflow of the electrolyte from the DAE module or the dry-up of the wetted foam. When glass foam is chosen as the porous media, quartz wool is tightly packed in between the foam and the electrodes to ensure the connectivity of the aqueous phase (Supplementary Fig. 7). The porous media also ensure the free movement of the electrolyte in the capillary of the foam (Supplementary Fig. 8, Supplementary Movie 1). The foam filled with ionic solutions forms a physical barrier that effectively isolates hydrogen, oxygen, and air from any mixing.

Hygroscopic substances characterized with a strong affinity with water tend to extract moisture from the atmosphere at exposure, absorbing sufficient water to form an aqueous solution which is hygroscopic in nature. When the chemical potential (μ) of water vapor

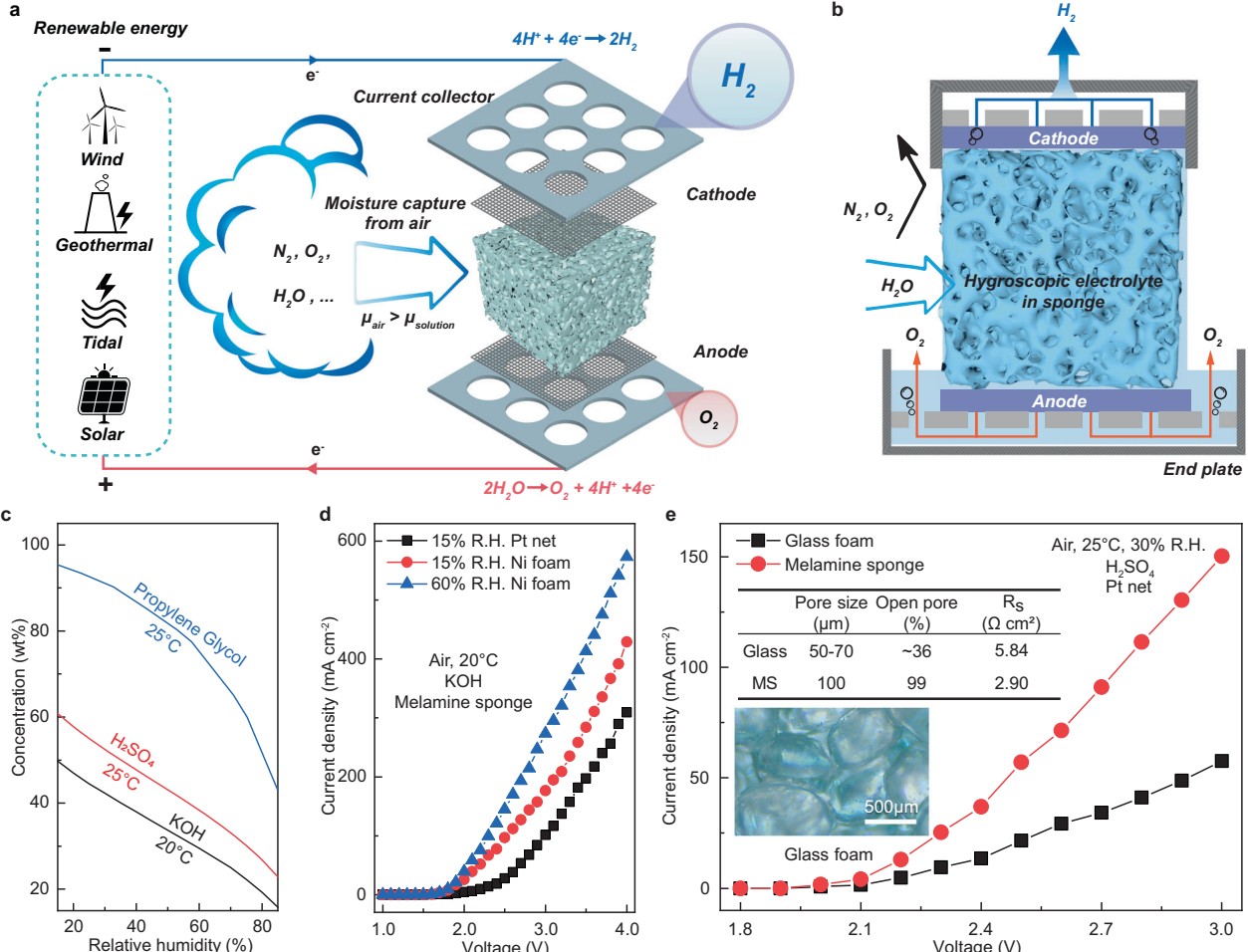

**Fig. 2 | The concept of direct air electrolysis (DAE) for hydrogen production. a** A schematic diagram of the DAE module with a water harvesting unit made of porous medium soaked with the hygroscopic ionic solution. **b** A schematic diagram of the cross-section of the DAE module, showing the electrodes are isolated from the air feed, and the absorbed water are transported to the electrode by capillaries of the sponge. **c** Equilibrium water uptakes of hygroscopic solutions at different air R.H.[31,32]. **d** J–V curves for DAE modules using Pt or Ni electrodes sandwiched with KOH electrolyte (in equilibrium with 15% and 60% R.H. at 20 °C) soaked in a melamine sponge. **e** Effect of sponge materials on J-V performance of DAE modules using $H_2SO_4$ electrolyte in equilibrium with 30% R.H. at 25 °C. The inset shows the optical micro image for the glass foam. Source data are provided as a Source Data file.

in the atmosphere is greater than that in a hygroscopic solution, i.e., $\mu_{air} > \mu_{solution}$, the solution will continue absorbing water vapor and being diluted until the vapor-liquid equilibrium is reached at $\mu_{air} = \mu_{solution}$[33], making the concentration of the solution C equal to the equilibrium one C* (Fig. 2c). In this study, we tested several hygroscopic materials, including $CH_3COOK$, KOH, and $H_2SO_4$, representing a salt, a base, and an acid, respectively. All three materials spontaneously absorb moisture from the air and form ionic electrolytes. It was found that the direct air electrolysis modules using the respective electrolytes were able to produce hydrogen gases successfully for an extended period with a continual supply of air and power. For $CH_3COOK$ based DAE module, the voltage was as high as 3.70 V due to the large size of acetate anions and substantial $CO_2$ and ethane byproducts found along with $O_2$ at the anode (Supplementary Fig. 9).

Importantly, the DAE module using Ni electrode (Supplementary Fig. 10) and KOH electrolyte with moisture supplied by 60% R.H. air achieved a high current density of 273 mA cm$^{-2}$ at 3.0 V and 574 mA cm$^{-2}$ at 4.0 V, or 177 mA cm$^{-2}$ at 3.0 V and 15% R.H. (Fig. 2d). However, the performance of this DAE module started to decline after 72 hr and we had to stop it at 96 h. This was because the voltage of the DAE module increased from 2.3 V to 2.4 V due to the gradual conversion of KOH into $K_2CO_3$ and eventually $KHCO_3$ at exposure to the 420 ppm

level $CO_2$ in the air. $KHCO_3$ is less soluble in water hence less conductive as an electrolyte, and critically it is non-deliquescent, unable to absorb moisture from the air. We believe if the $CO_2$ in the feed air can be rejected by a barrier, KOH would stay as a top choice for the DAE module.

Sulfuric acid has been identified as one of the best hygroscopic materials that can absorb moisture from the air down to relative humidity 5% or below[31,34]. Meanwhile, the sulfuric acid solutions are high in conductivity (0.61 S cm$^{-1}$ at 50.0 wt%)[35,36], non-volatile, and it is non-toxic to the environment. It was found the current density of the DAE using $H_2SO_4$ soaked melamine sponge could also reach 150 mA cm$^{-2}$, 2.5 times higher than that using sintered glass foam, because the series resistance of the former was 50% lower owning to its high open-pore fraction (Fig. 2e). However, the melamine sponge gradually degrades in the $H_2SO_4$ solution after a week. In this regard, the following studies were carried out using sulfuric acid electrolyte equipped with glass foams and platinum (Pt) mesh electrodes (Supplementary Fig. 11) for long-term stability and $CO_2$ resistance. It is also interesting to note that in the concentration range of sulfuric acid of this work, the corresponding freezing point of the electrolyte is below −30 °C[37], implying potential working temperature under an icing environment.

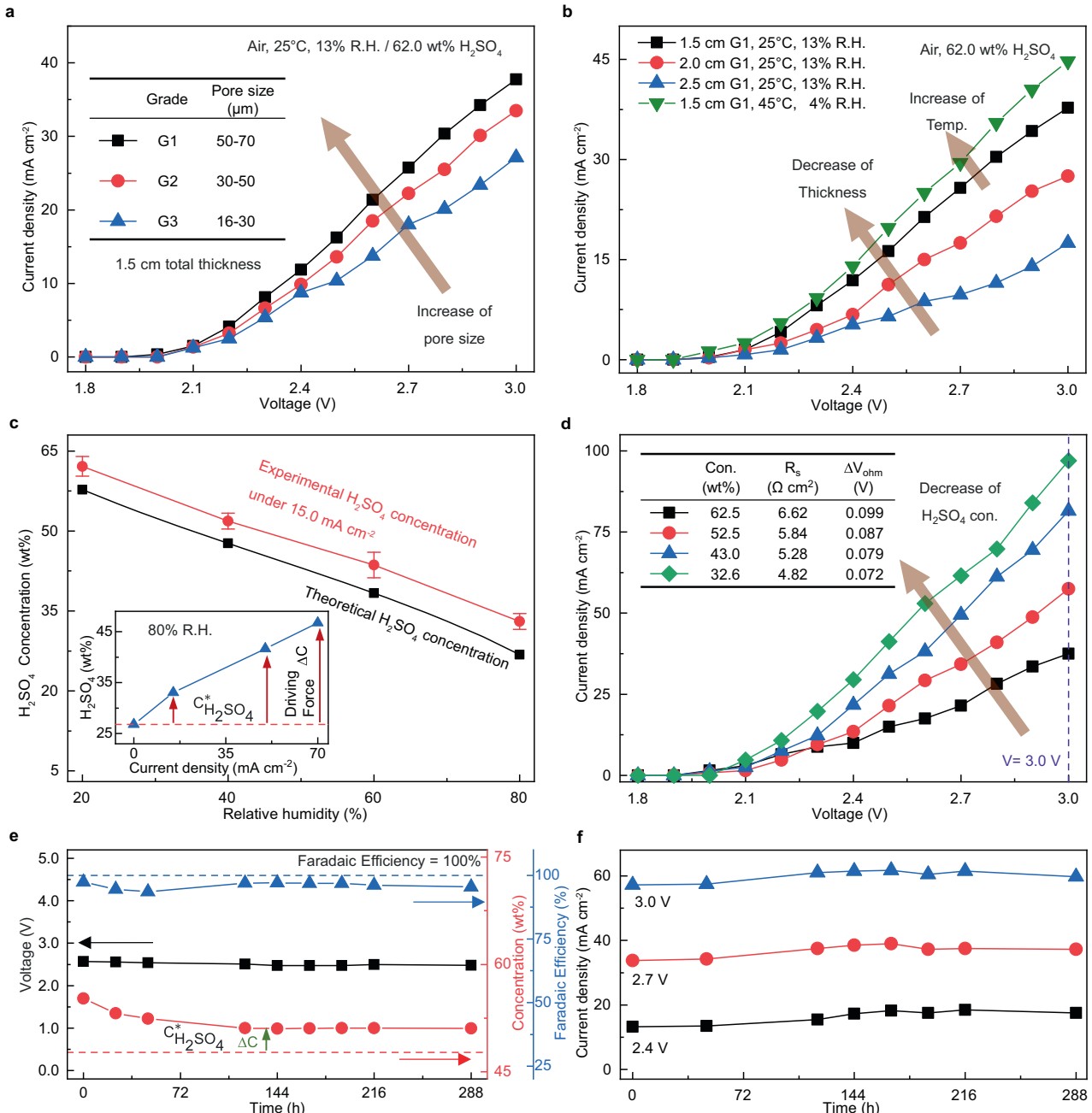

**Fig. 3 | DAE module performance at 25 °C unless specified. a** J–V curves for modules using various porous substrate with different pore sizes. **b** Effect of temperature and foam thickness on J–V performance. **c** The experimental steady state concentration of sulfuric acid at J = 15.0 mA cm⁻² (red) VS equilibrium concentration $C_{H_2SO_4}^*$ (black) under different R.H. Error bars represent standard deviation, n = 3 independent replicates. The inset shows the effect of current density on steady state concentration under 80% R.H. where $C_{H_2SO_4}^* = 26.8$ wt% (red

dashed line). **d** The effect of electrolyte concentration on J–V performance. **e** Example recording of cell voltage (black symbol), $H_2SO_4$ concentration (red symbol),$C_{H_2SO_4}^* = 47.7$ wt% (red dashed line), Faradaic efficiency (blue symbol) for DAE modules at constant current density of 15.0 mA cm⁻² for 288 h at 40% R.H. **f** Current density collected under different voltage for 288 h at 40% R.H. Figures 3c to 3f were operated with G1 sintered glass foam and 1.5 cm gap between the two electrodes. Source data are provided as a Source Data file.

## Performance of the DAE module

The DAE module's performance was investigated using current density (J) and voltage (V) characteristic experiments conducted at 25 °C. The effect of relative humidity ranging from 20% to 80%, as well as the pore size and thickness of the sintered glass foams, were studied. Sintered glass foams were labelled as G1, G2, and G3 corresponding to the pore size of 50−70, 30 − 50, 16−30 μm, respectively. Also, a series of experiments with extended time durations of 288 h was conducted to investigate the stability of the DAE module.

The effect of the different pore sizes of sintered glass foams on the J-V behavior is shown in Fig.3a, using 62.0 wt% $H_2SO_4$ solution as electrolytes. Current density is negligible (<1 mA cm⁻²) at a voltage below 2.0 V due to the overpotential of the Pt mesh. As long as the capillary force still holds the electrolyte, the current density increases with the use of larger pored sintered glass foams, indicating higher conductivity and energy efficiency for overall water splitting due to better mobility of electrolyte in larger pores. At 3.0 V, a current density of 27.1 mA cm⁻² was achieved using G3 sintered glass foam and it

increased to 37.8 mA cm$^{-2}$ using G1 sintered glass foam. Therefore, the G1 sintered glass foam was chosen for further study of the foam thickness, owning to the high electrical conductivity, low resistance, and high energy efficiency it brings to the DAE module.

The operation temperature and sintered glass foam's thickness also plays a role in the J-V behavior. As shown in Fig.3b, with the increase of temperature from 25 °C to 45 °C, the current density for the DAE module increased from 37.8 mA cm$^{-2}$ to 44.8 mA cm$^{-2}$, under a constant voltage of 3.0 V. This can be attributed to the improved ion conductivity of H$_2$SO$_4$ with elevating the temperature[35]. In the meantime, the J-V curve shifts upwards with decreasing glass foam thickness at 25 °C. At 3.0 V, the current densities are 17.5 and 37.8 mA cm$^{-2}$ while using 2.5 and 1.5 cm thickness G1 sintered glass foams, respectively. According to Pouillet's law[38], the resistance is proportional to the distance between the electrodes, suggesting that a large distance between the cathode and anode contributed to high resistance for overall water splitting. Hence, under specific current density, the gap between two electrodes should be as small as possible to maintain relatively high energy efficiency. However, the mass transfer area for water absorption is proportional to the sintered glass foam's thickness. There is a trade-off between the water absorption area and conductivity. Considering both factors, we chose the G1 sintered glass foam with 1.5 cm total thickness for further investigation, given that it could provide sufficient mass transfer area for air-electrolyte contact while maintaining moderate energy efficiency.

The observed experimental concentration of sulfuric acid C is constantly above its equilibrium concentration C* during the direct air electrolysis process. This difference represents the driving force for the mass transfer of water from the vapor phase into electrolyte solution and then onto the electrochemical reaction sites at the electrodes. Figure 3c shows that at J = 15.0 mA cm$^{-2}$, the experimental concentration in the DAE module is approximately 5 wt% higher than the equilibrium at steady state, which means a stable in situ H$_2$SO$_4$ concentration over 8 h under a constant current density, where the rate of water absorption from air equals the rate of water consumption by electrolysis. Likewise, such steady-state mass transfer driving force can be established at fixed air relative humidity. As shown in Fig. 3c inset, the driving force increases proportionally with the increase of current density, which means the rate of water absorbed by the DAE module rises when the water electrolysis rate is turned up. For instance, at R.H. = 80%, if a minimal current density is applied, the sulfuric acid concentration in the module is close to the equilibrium C*$_{H_2SO_4}$ = 26.8 wt%, and the mass transfer driving force of water absorption is nearly zero. If we increase the current density J to 70 mA cm$^{-2}$, the steady-state concentration of sulfuric acid is increased to 46.7 wt%, 75% higher than the equilibrium one C*$_{H_2SO_4}$ = 26.8 wt%. Therefore, our DAE module is intrinsically self-converged, compatible with a broad range of air humidity and current density.

The DAE module's J-V behavior has also been studied under different H$_2$SO$_4$ concentrations (Fig.3d). With the decrease of H$_2$SO$_4$ concentration from 62.5 wt% to 32.6 wt%, the series resistance of the system decreases from 6.62 Ω cm$^2$ to 4.82 Ω cm$^2$, while the current density for the electrolysis reaction increases significantly from 37.5 mA cm$^{-2}$ to 97.0 mA cm$^{-2}$, under a constant voltage of 3.0 V (the iR-corrected J-V curve is shown in Supplementary Fig. 12). Such change can be attributed to the improved electrical conductivity of diluted H$_2$SO$_4$ (Supplementary Fig. 13)[35,36]. Also, the viscosity of the electrolyte decreases as the acid is being diluted (Supplementary Fig. 14), resulting in higher electrocatalytic activity and reduced electrochemical polarization[39,40]. It is worth comparing the DAE using H$_2$SO$_4$@sintered glass with that of KOH@melamine sponge, the latter has a system series resistance of 2.93 Ω cm$^2$, only 0.20 Ω cm$^2$ higher than an electrolyzer using direct KOH solution i.e., foam free electrolyzer (Supplementary Fig. 15). Such a low series resistance is responsible for the high current density of 574 mA cm$^{-2}$ achieved by the DAE module using KOH@melamine sponge at 4.0 V as mentioned earlier.

The DAE module was found stable during continual electrolysis. Performance of the electrolysis cell at various voltage, energy efficiency, and air R.H. are shown in Supplementary Table 1 and Supplementary Fig. 16. After a minor fluctuation initially, the J-V behavior stabilize for a 48 h run. For further laboratory test, we chose 40% R.H. air as the gas atmosphere condition. As shown in Fig.3e, the concentration of H$_2$SO$_4$ fed to the module was 55.2 wt% initially, and it converged to 51.1 wt% over the first 120 h. In the following 168 h, the electrolyte concentration, the DAE module's voltage, the mass transfer driving force for moisture absorption (ΔC = C$_{exp}$[51.1 wt%]−C*[47.7 wt%] = 3.4 wt%) and the H$_2$ Faradaic efficiency (around 95%) are all stabilized. Accordingly, the corresponding current densities collected under specific voltages (2.4, 2.7, 3.0 V) also reached steady state in this 12-day continual operation (Fig.3f). This result indicates excellent adaptability and long-term stability of the DAE modules under different air R.H., cell voltage, and electrolyte concentrations.

## Demonstration of DAE modules stack with solar panel in the open air

To further demonstrate the DAE module's working capability in a practical environment, we designed and constructed a free-standing hydrogen generation tower consisting of five DAE modules stacked in parallel superimposed vertically with a solar panel for power supply. The details of the structure of the tower are shown in Fig.4a and Supplementary Fig. 17. One of the advantages of such design is that the footprint of the tower is no more than the solar panel, meaning our DAE will not occupy extra land, which can be costly in some areas. The tower was tested for two days, 8 h per day, in the open air of a hot-dry summer (Mediterranean climate) in the campus of the University of Melbourne. The setup of the module of the outdoor surroundings is provided in Supplementary Movie 2. The outdoor temperature varied from 20 °C to 40 °C, and the relative humidity ranged from 20% to 40% over the testing period. Since the solar panel was used as a renewable energy supply, the voltage, and the current of each DAE module were solely determined by solar intensity (Supplementary Movie 3), which varies every hour. The product hydrogen gas evolved from the cathode was collected in an inverted, water-filled cylinder over water, which was then used to examine the gas production rate (Supplementary Movie 4). The oxygen generated on the anode of the DAE was vented into the air.

The performance of the tower was shown in Fig.4b and Supplementary Fig. 18, in the form of the hydrogen generation rate, hydrogen evolution Faradaic efficiency ($\eta_{f,H_2}$), the overall current, and the voltage. During the open-air demonstration, $\eta_{f,H_2}$ was at an average of 95% during the daytime, shown as the red line in Fig.4b. On the first day, when the weather was sunny, the current output was stable around 400 mA, and voltage 2.68 V. The hydrogen evolution rate was 186 ml h$^{-1}$, with the total hydrogen production at 1490 ml in a day, which is equivalent to 745 L H$_2$ day$^{-1}$ m$^{-2}$ of the cathode, or 3.7 m$^3$ H$_2$ day$^{-1}$ (m$^2$ tower)$^{-1}$.

On the second day, a few hours of good sunlight guaranteed the current output stable at 400 mA from 9:00 to 13:00, with an average hydrogen generation rate of about 179 ml h$^{-1}$, similar to that of the first day. However, in the early morning from 8:00 to 9:00, the solar intensity was limited, resulting in a relatively lower current output of 270−370 mA and a hydrogen generation rate of 140 ml h$^{-1}$. In the cloudy late afternoon (14:00 to 16:00), the poor weather conditions reduced the solar panel's current output to as low as 50 mA, and hence, the hydrogen generation rate dropped to 21 ml h$^{-1}$. On the whole, under non-ideal weather conditions, the total hydrogen production could still reach 1188 ml on the second day.

The gas product collected from the cathode has been analyzed with gas chromatography (GC.), suggesting pure hydrogen (>99%) (Supplementary Fig. 19). The gas produced from the anode has also been measured (Supplementary Fig. 20) with a GC. showing it is a high-

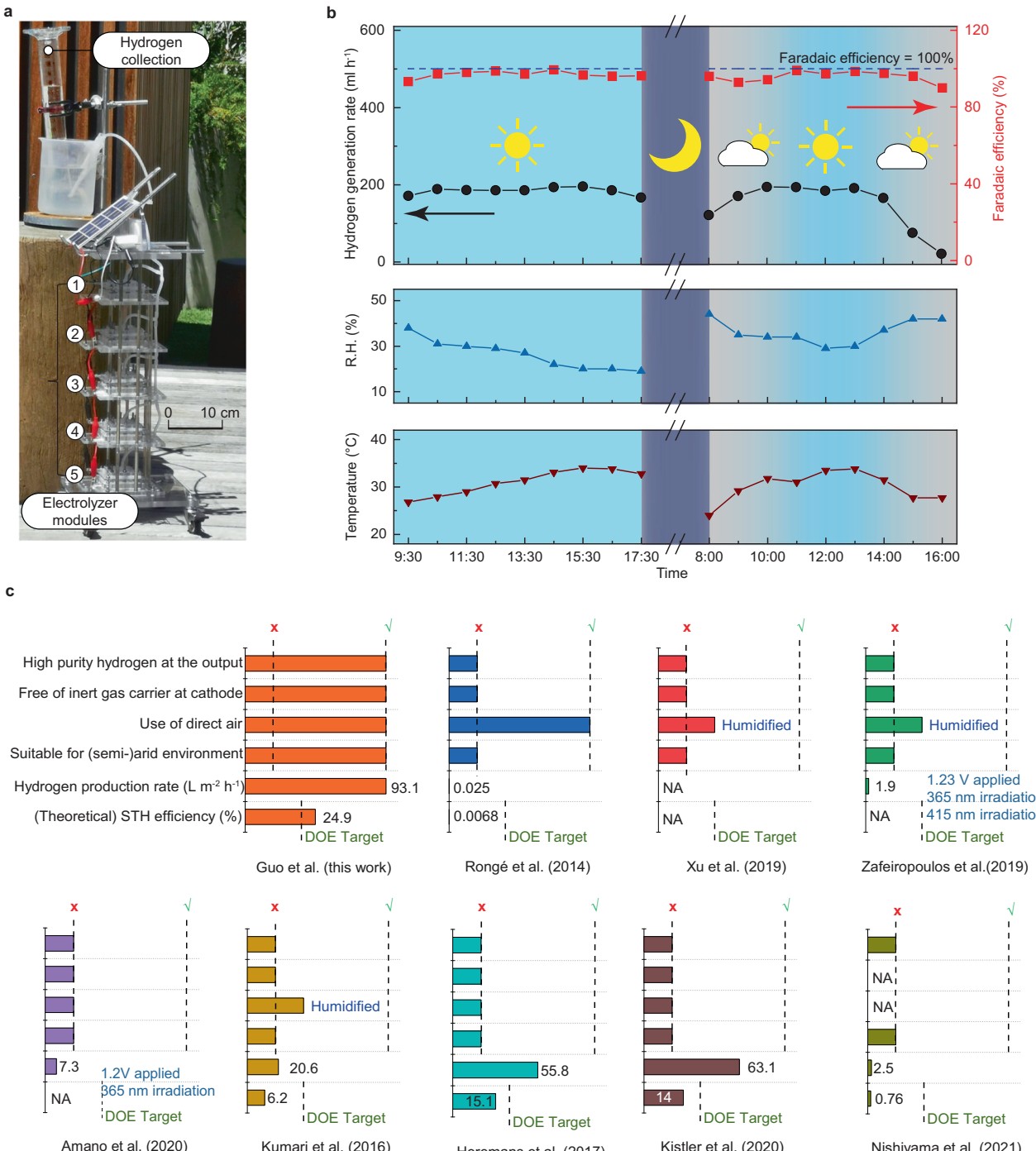

**Fig. 4 | Open-air demonstration of the hydrogen generation tower in Dec 2020 Melbourne. a** Photo of the designed hydrogen generation tower featuring scalable DAE modules with minimized footprint. **b** Hydrogen generation rate and Faradaic efficiency on an hourly basis at corresponding air humidity and temperature for two days under different weather conditions. Source data are provided as a Source Data file. **c** Comparisons between our work and the top performing solar-driven water/vapor splitting reported by others (see Supplementary Table 2 for more details)[14–20,22].

purity oxygen (>99%) (Supplementary Fig. 21). The Faradaic efficiency of oxygen $\eta_{f,O_2}$ at the anode is over 91.0%. Since the Faradaic efficiency of both $H_2$ and $O_2$ measured and calculated by energy and mass balances are comparable, we again confirm the overall electrolysis is a water-splitting process. After keeping the above DAE module unattended in air for 8 months, the Faradaic efficiency of hydrogen remains around 90%, without any maintenance.

We compared the DAE modules against the top-performing solar-driven water/vapor splitting by others. The top criterion is the ability to produce high-purity hydrogen. As shown in Fig.4c, apart from our

work, the existing electrolyzers using vapor feed or photocatalytic water splitting have not been able to deliver >99% purity hydrogen at the cathode output with their current process setup. Normally, vapor-fed electrolyzer can only produce $H_2$ with no more than 2% purity[14–18], e.g., works by Kumari et al[17], and Kistler et al[18]; whereas photocatalysts produce a mixture of $H_2/O_2$. It must be emphasized that all the literature using vapor feed (at the anode) need inert carrier gases ($N_2$ or Ar) at the cathode which explains why their product $H_2$ purity was so low. Frequently, for those using vapor feed to the anode, artificial humidification has been employed to boost the humidity of the feed to

above 60%[14–20], making them unsuitable for (semi-)arid environment. Also, the hydrogen production rates in the existing vapor-fed electrolyzers are mostly lower than 65 L m$^{-2}$ h$^{-1}$, while our DAE prototype can reach 93.1 L m$^{-2}$ h$^{-1}$ in the outdoor test (Fig.4c). In comparison, our DAE module is the only one that ticks all the above boxes, plus a demonstrated 8 months long term stability to-date, and a hydrogen production rate substantially (from a few times to a few orders of magnitude) higher than literature works in the same domain[14–19,22]. Very importantly, to the best of our knowledge, our DAE could be the first technology exceeding the target of 20% solar-to-hydrogen (STH) energy efficiency set by the U.S. Department of Energy (DOE)[41]. DAE coupled with a triple-junction solar panel can achieve a theoretical STH efficiency of 15.7% under different H$_2$SO$_4$ concentration (Supplementary Fig. 22), while coupling with the best performing solar panel using H$_2$SO$_4$ and KOH hygroscopic electrolyte can achieve a theoretical STH efficiency of 24.9% and 32%, respectively (see Supplementary Tables 1, 3 and 4 for more details).

Furthermore, the energy supply to our DAE module is not limited to solar. In another demonstration (Supplementary Fig. 23 and Supplementary Movie 5), we coupled the DAE module with a wind turbine and successfully produced high-purity hydrogen from the air feed.

## Discussion

In summary, to overcome the water shortage problem in the case of widespread deployment of hydrogen production, we have demonstrated a method of producing high purity hydrogen from the air by using hygroscopic electrolyte soaked in a porous medium as the moisture absorbent. Our direct air electrolysis (DAE) module can achieve exceptional performance under specific conditions, such as operational at as low as 4% relative humidity with H$_2$SO$_4$ hygroscopic electrolyte, or more than 12 days continuous H$_2$ generation at 40% relative humidity performing at a hydrogen Faradaic efficiency around 95% without any decay or attendance; while in the case of using KOH hygroscopic electrolyte and nickel foam electrodes, the current density can reach 574 mA cm$^{-2}$ at 4.0 V and 60% R.H., or 177 mA cm$^{-2}$ at 3.0 V and 15% R.H. This DAE module can be numbered up easily and integrated with various renewable powers. In our demonstration, a prototype of standalone hydrogen generation tower with five DAE modules stacked in parallel superimposed vertically under a solar panel (with minimum footprint) was constructed and tested outdoor in a hot-dry summer. The prototype achieved automated steady hydrogen production depending on the solar intensity under varying weather conditions. On a warm sunny day, the hydrogen production rate can reach 3.7 m$^3$ H$_2$ day$^{-1}$ m$^{-2}$ tower. The integration of the DAE with wind turbine was demonstrated in this study as well. Such DAE farms hold the potential for generating abundant hydrogen in arid and semi-arid areas with negligible disrupt to the regional air humidity and minimal impact to the environment (Supplementary Note 1). Further improvement of the surface-to-volume ratio by engineering channels or increasing the aspect ratios of the sponge material will guarantee the rate of water uptake which is essential to the upscaling of the DAE units.

## Methods

### DAE module fabrication

A self-designed DAE module was used under each R.H. in this project. More details can be found in the Supplementary document. Pt mesh electrodes (99.99% purity, made by 0.12 mm Pt wire, while the frame was 0.5 mm Pt wire, Yueci technology Co.) or Ni Foam electrodes (1.6 mm thickness, Keshenghe metal materials Co.) with geometric area 4 cm$^2$ were attached directly to the quartz wool (99.95% purity, 5–10 μm, Xinhu Co.), and then connected with the sintered glass surfaces, with geometric area 7.84 cm$^2$ (Shundao sintered glass foam Co.). The melamine sponge (Daiso Industries Co., Ltd.) could replace the glass foam and the quartz wool, with the geometric area 7.84 cm$^2$.

The sintered glass foams' thickness was 3 mm, and quartz wool was layered between two foams. For example, for 1.5 cm total thickness, three foams and four layers of quartz wool were used stacked. Teflon plate with Pt wireline (99.99% purity, Xudong Co., Ltd.) was used as current feeders and electrolyte distributors. After assembly, the DAE module was put inside the climate test chamber (DHT-100-40-P-SD, Shanghai Doaho Co., Ltd.), keeping a close environment at a constant R.H. and temperature. The DAE module connected directly with a D.C. power supply (DPS3010U, Wanptek Co.), which could supply constant current for electrolysis. The cathode's output gas production was bubbled through a water bath, and collected in an inverted, liquid-filled cylinder.

### Data collection

Another DC power supply (Nice Power R-SPS605D, ShenZhen Kuaiqu electronic Co., Ltd.) was used to connect with the DAE module or foam-free electrolyzer to collect the current density vs. voltage (J–V) performance curve. The foam-free electrolyzer was a 50 ml volume two-electrode cell, with a 1.5 cm electrode distance. The current was measured after 30 s under each voltage, using an applied voltage from 1.80 V to 3.00 V (H$_2$SO$_4$) or 1.00 V to 4.00 V (KOH) with a 100 mV increase per 30 s. The area of the electrode was 4 cm$^2$. Under each R.H., the J-V behavior was tested before putting into the environment oven and after operating over 24 h and 48 h. Each J-V behavior was verified by repeated measurements three times, with the current densities variation controlled within 5–10%.

Under each R.H., the DAE module was operated under constant current density, and the weight needed to be checked each 4 h until it maintained stable over an 8-hr period. The electrolyte equilibrium concentration was calculated by the weight changes of the DAE module before operation and after steady state.

Electrochemical impedance spectroscopy (EIS) measurements were employed at 0 V vs (OCP) with the frequency range from $10^6$ Hz to $10^{-1}$ Hz and an AC signal of 10 mV in amplitude as the perturbation for collecting the series resistance (CS350 Electrochemical Workstation, Wuhan Corrtest Instrument Co., Ltd.).

In this paper, the iR-compensation was calculated by the following equation:

$$V_{iR-compensation}(V) = V_{original}(V) - J(mA\,cm^{-2}) \times R_s(\Omega\,cm^2) \times 10^{-3}(A\,mA^{-1}) \quad (1)$$

where voltage $V_{original}(V)$ and current density $J(mA\,cm^{-2})$ were collected from the J-V curve, and $R_s$ ($\Omega\,cm^2$) was series resistance for each DAE module.

### Faradaic efficiency

The gas product flowed into a measuring cylinder (25.0 ml) through a rubber pipeline for volume measurement by a collection of gases in an inverted, water-filled cylinder over water. Gas collected inside the cylinder was drawn out by the syringe and then pushed into the gas chromatography (G.C.) system (7890B, Agilent technologies, Inc.) with a thermal conductivity detector (TCD.) for analyzation. The separation columns used in the GC were HP-INNOWAx, HP-PLOT U, and CP-Molsieve 5 Å Columns. The Faradaic efficiencies $\eta_{f,H_2}$ and $\eta_{f,O_2}$ were compared to the gas production with the ideal production rate, which is calculated according to the following equations:

$$FE_{H_2} = \frac{r_{H_2}}{r_{H_2,ideal}} = \frac{r_{H_2}(ml\,s^{-1})}{\frac{J(mA\,cm^{-2}) \times S(cm^2) \times 10^{-3}(A\,mA^{-1})}{2F} \times \frac{RT}{P_0} \times 10^6(ml\,m^{-3})} \quad (2)$$

$$FE_{O_2} = \frac{r_{O_2}}{r_{O_2,ideal}} = \frac{r_{O_2}(ml\,s^{-1})}{\frac{J(mA\,cm^{-2}) \times S(cm^2) \times 10^{-3}(A\,mA^{-1})}{4F} \times \frac{RT}{P_0} \times 10^6(ml\,m^{-3})} \quad (3)$$

where $r_{H_2}$ and $r_{O_2}$ were the rate of hydrogen production and oxygen generation rate respectively, while $r_{H_2,ideal}$ and $r_{O_2,ideal}$ were the ideal rate of hydrogen production and oxygen generation rate respectively. $P_O$ was the standard atmospheric pressure (101,325 Pa), T was the operating temperature (298.15 K), R was the gas constant (8.3145 m$^3$ Pa K$^{-1}$ mol$^{-1}$), F was Faradaic constant (96,485 C mol$^{-1}$) and S was the electrode area (4 cm$^2$).

## Materials characterization

The FlexSEM 1000 scanning electron microscope (SEM, Hitachi Co., Ltd.) and Binocular microscope BM-500T (Ruihong Co., Ltd.) were used for imaging the glass foam.

## Stability test

55.2 wt% $H_2SO_4$ (formulated with 98% sulfuric acid and R.O. water) was used as the electrolyte and tested under 40% R.H. and 25 °C in the climate test chamber for 12 days under constant current density 15.0 mA cm$^{-2}$.

## Open air demonstration with solar panel

Five DAE modules were stacked vertically in parallel, and the hydrogen generation was collected. Supporters were used to keep the distance between each unit. Here, a commercial silicon solar panel was connected in series and put on the top of the units, with an open-circuit voltage of around 6.0 V and a short circuit current around 400 mA under Melbourne's natural sunlight. The gas product was also flowed into a measuring cylinder by collecting gases in an inverted, water-filled cylinder over water. Also, the gas product could be collected in an inverted, oil-filled cylinder over oil.

## Open air demonstration with wind turbine

Here, a commercial small wind turbine was coupled with a single DAE module, with an open-circuit voltage of around 8.0 V. However, short circuit current was very low (<1 mA). The gas product could be bubbled in oil.

## Data availability

Source data are provided with this paper.

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

## Acknowledgements

J. G. is grateful for the Melbourne-Manchester Graduate Research scholarship.

This work was performed in part at the Materials Characterization and Fabrication Platform (MCFP) at the University of Melbourne and the Victorian Node of the Australian National Fabrication Facility (ANFF).

## Author contributions

G.L. conceived the idea. J.G. and Y.Z. conducted the experiments. J.G., G.L., A.Z., and Y.Z. analyzed the data. K.C. and Y.G. helped with characterizations. J.G. led the draft of the manuscript with input from G.L., A.Z., X.F., G.H., and Y.Z. This project has been supervised by G.L., G.H., and X.F.

## Competing interests

The authors declare no competing interests.
