## [Peer Review File · Nature Communications]

Hydrogen Production from the AirEditorial Note: This manuscript has been previously reviewed at another journal that is not operating a transparent peer review scheme. This document only contains reviewer comments and rebuttal letters for versions considered at *Nature Communications*. Parts of this Peer Review File have been redacted as indicated to maintain the confidentiality of unpublished data.

REVIEWER COMMENTS

Reviewer #2 (Remarks to the Author):

My concerns have been addressed by the revisions.

Reviewer #3 (Remarks to the Author):

The authors describe an elegant air-based hydrogen production device that can operate in harsh environments with low water activity typical of high-solar areas. In particular, the strength of this device lies in its ability to utilize the water vapor at low RH and the air-separated cathode compartment facilitates the production of high purity hydrogen. Below you will find my comments which need to be addressed. In addition, I was asked to provide my thoughts on the comments and the replies given concerning the remarks of referee #1.

Comments:

a) In the abstract, the authors mention that 'The DAE modules can be easily scaled to provide H₂ to remote, (semi-)arid, and scattered areas.' The surface area at the sides of the sponge/foam is used for water vapor capturing. This area is small but sufficient for a small device. I would assume this may become an issue for a large device because the increase in surface area for water uptake may not be sufficient anymore to replenish the larger volume of depleted water content. Therefore, it is too early to make this claim. Providing a comment about this in the manuscript on how this can be addressed in future work may be of value.

b) The captured water is said to be transported by diffusion in the sponge/foam. Is this mass transfer phenomenon fast enough to replenish the water inside the foam to keep the air from entering the cathode compartment through the foam, or is capillary action created by the water reservoir in the anode compartment the main contributor to water flow? Currently, this reservoir is used as an air barrier and to store excess liquid. It would be good to know if the system can still operate in a stable way without this additional water reservoir in the anode compartment. The presence and functionality of this reservoir should be clearly mentioned in the main text if it is really that important for the functioning of the system.

c) In 'Supplementary Note 1' (SI) a height of 30m is used in eq12 which results in only a slight decrease in RH. The 30m feels handpicked to reach the desired results. The analysis should be done more carefully.

d) Spelling mistake in Fig.1a (Solar potential)

e) Sulfuric acid is hygroscopic but I'm not sure if it can be classified as deliquescent

f) In SI: Compared with Fig.S20a and Fig.S20b, we can confirm that the gas production at the cathode is high purity O₂(>99%). This should be the anode.

I added my comments to the summarized report of referee #1 below.

The authors claim that their work is unique because a) it can be powered by any renewable energy source; b) it is the only one that can use direct air feed to cathode and anode; c) it is the only one that can produce high purity hydrogen; d) it is the only one that works in dry environment and e) it is the only one having a STH efficiency above 20%.

Point a) is unconvincing, since any air electrolyzer can be operated with any source of power, e.g. the devices reported by Kumari, Heremans and Kistler.

Point a is addressed by the authors.

Point b) is not correct. The device reported here feeds air to the glass separator, not to the anode or cathode directly. It is also not clear why this would be a direct advantage. It could be indirectly beneficial for the purity of the hydrogen (see next point).

Point b is addressed by the authors.

Point c) is not entirely correct. While it is true that most reports in the past have used a flow of (typically) inert gas at the cathode, this served mostly analytical purposes (efficient evacuation of the gas for product analysis). Note that e.g. Xu et al (2019) utilize a dry gas stream at the cathode not supplying any water vapor. There is no fundamental reason why these reported devices couldn't operate without a continuous gas flow at the cathode, thus resulting in high purity hydrogen. Rongé et al. (2014) indeed did not use a continuous gas flow at the cathode, simply an initial purge. Kistler et al. (doi: 10.1149/2.0041905jes) performed experiments without any gas flow at the cathode. Thus, while it is true that most earlier devices utilized a gas flow at the cathode, one cannot conclude that it is impossible to obtain a high purity hydrogen product with those devices.

There seems to be some disagreement between the authors and referee#1 on this point so I will try to elucidate both standpoints to try to come to a resolution. In their response, the authors have demonstrated that it is not possible for the devices from the literature to reach high H₂ purity in the way these were operated. This was also demonstrated by the actual purity of the produced H₂. It looks to me that reviewer #1 has a problem with the wording of the statements in the main text but not necessarily with the actual argument. To give an example from the text: 'none of the existing electrolyzers using vapor feed or photocatalytic water splitting can deliver >99% purity hydrogen at the cathode output'. I would say that we cannot know with certainty that these devices cannot reach high purities if they are operated in a different process setup. It would be safer to say that these devices have not been able to reach high purities with their current process setup.

Point d) is justified and is the main benefit of the device reported here. While others have reported good performance at RH down to 30%, this device has the potential to operate better than earlier devices at even lower humidities.

Point d is addressed by the authors.

Point e) is utterly misleading. Some of the earlier devices operated at electrolyzer efficiencies well above 50%, while the electrolyzer reported here achieves an energy efficiency of 43.8-52.8%. When combined with a conventional silicon solar cell, this would result in STH efficiency well below 20% (as the authors admit themselves in Supplementary Table 4). The number of 32% which is quoted in the main text is purely theoretical and based on a combination with an exotic PV cell at extremely high efficiency. The same theoretical argument could be made for any of the devices already reported in the literature. Moreover, it neglects the impact of coupling efficiency. Which brings me to the following point: one should only speak of STH efficiency if one has actually done the experiment and can show the data. The authors have made a demonstration with a conventional solar cell and should use these data to claim any STH efficiency in the text (which can only be done if they have measured the solar irradiance).

The authors gave a reasonable alternative definition for the theoretical STH. Because this STH value depends on the efficiency of the solar cells, it is not a conclusive indication that the DAE device performs better than other comparable devices. It would be better to compare the efficiencies of the electrolyzers. The electrolyzer constructed by the authors provides a reasonable efficiency but is not the best in the field. This is ok as the main selling point would be the air-based concept at low RH with high H₂ purity. And more importantly, I'd say it would be intuitively normal to achieve lower

performance at low water activity. A comparison between performances of other devices found in the literature (e.g. current density at a certain operating voltage, electrolyzer efficiency,...) specified at a specific RH can be valuable to the reader and this would provide a more fair comparison in terms of efficiency/performance.

The authors often refer to Faradaic efficiency. However, the GC results for experiments with sulfuric acid should be added to the supplementary material (they are currently only available to referees). Moreover, this represents but one measurement, while the authors claim certain numbers for FE during 12 days and even 8 months. Such claims are irrelevant if they are not backed by data. Fig. 3 shows the operating voltage in an experiment at (very low) constant current during 12 days. FE data should be added to these graphs. Similarly, such data should be made available for the 8-month experiment as well.

The authors addressed these comments partially. Referee #1 asked for the FE data in figure 3, can this be provided?

The authors' claims are often an act of cherry picking: STH efficiency is calculated based on a cell with KOH (and with H₂SO₄) at low current density, high RH and with an exotic, hypothetical PV device. The stability is based on experiments at very low current density (15 mA/cm²) with sulfuric acid. The high current density of 574 mA/cm² quoted in the abstract is based on an unstable system with KOH and melamine separator, at 60% RH. Meanwhile, they also claim to operate at RH as low as 4%, which is demonstrated only at elevated temperature of 45 °C and reaching current densities around 30-45 mA/cm² at electrolyzer energy efficiency of 40-45%. Such claims are misleading and distracting. The merit of this work is the demonstration of a device based on sulfuric acid, noble metals and porous glass, which can continuously capture and split water from air even at low humidity. It is, however, not groundbreaking nor better performing than the multitude of devices presented until now.

Indeed, the performance of the device is not better than other related devices but the high purity hydrogen and the ability to operate at extremely low RH conditions is certainly an important characteristic. With the newly introduced concepts of using air as a water source, the ways of comparing data must also evolve e.g. the RH is likely another parameter that needs to be taken into account. However, it is necessary to address the other points mentioned in the above comment of referee #1. More reasonable numbers should be quoted for the actual well-performing and stable devices.

06/06/2022

Response to Reviewers

The following text in **blue** are the questions raised by reviewers and our responses are in black, and the corresponding revisions quoted from the manuscript in **red**.

Reviewer #2:

My concerns have been addressed by the revisions.

RE: We thank the reviewer once again for the positive and conclusive comments.

Reviewer #3:

The authors describe an elegant air-based hydrogen production device that can operate in harsh environments with low water activity typical of high-solar areas. In particular, the strength of this device lies in its ability to utilize the water vapor at low R.H. and the air-separated cathode compartment facilitates the production of high purity hydrogen. Below you will find my comments which need to be addressed. In addition, I was asked to provide my thoughts on the comments and the replies given concerning the remarks of referee #1.

Comments:

a) In the abstract, the authors mention that 'The DAE modules can be easily scaled to provide H₂ to remote, (semi-)arid, and scattered areas.' The surface area at the sides of the sponge/foam is used for water vapor capturing. This area is small but sufficient for a small device. I would assume this may become an issue for a large device because the increase in surface area for water uptake may not be sufficient anymore to replenish the larger volume of depleted water content. Therefore, it is too early to make this claim. Providing a comment about this in the manuscript on how this can be addressed in future work may be of value.

RE: We thank the reviewer for the positive views on our manuscript and the thoughtful comments that helped us enhance our work's quality.

The reviewer raised a good question on the water uptake – such question has been asked by venture capitalists working with us. It is true that when our device of air electrolyzer is scaled up, the surface-to-volume ratio must not be reduced. In order to keep the water uptake per unit volume sufficient, one of the solutions is to design horizontal channels in the bulk body of the sponge/foam material, and the channels should be big enough, e.g. ¼” to ½” width, to allow for air passing

through, as shown in the Supporting Figure below. Such design will maintain the surface-to-volume ratio of the electrolyzer.

[redacted]

We have also incorporated your remark on page 14 of the revised manuscript:

Further improvement of the surface-to-volume ratio by engineering channels or increasing the aspect ratios of the sponge material will guarantee the rate of water uptake which is essential to

the upscaling of the DAE units.

b) The captured water is said to be transported by diffusion in the sponge/foam. Is this mass transfer phenomenon fast enough to replenish the water inside the foam to keep the air from entering the cathode compartment through the foam, or is capillary action created by the water reservoir in the anode compartment the main contributor to water flow? Currently, this reservoir is used as an air barrier and to store excess liquid. It would be good to know if the system can still operate in a stable way without this additional water reservoir in the anode compartment. The presence and functionality of this reservoir should be clearly mentioned in the main text if it is really that important for the functioning of the system.

RE: Thanks for the comments.

Yes, you are correct. Capillary force is responsible for replenish water inside the form that keeps air from entering the cathode. We actually mentioned “capillary” twice in our last manuscript: on page 4 “**The porous media also ensure the free movement of the electrolyte in the capillary of the foam (Fig.S8, Video S1). The foam filled with ionic solutions forms a physical barrier that effectively isolates hydrogen, oxygen, and air from any mixing.**” and on page 7 “**As long as the capillary force still holds the electrolyte, the current density increases with the use of larger pored sintered glass foams**”. On the other hand, “diffusion” is essential for the transport of the freshly captured water from the air-foam interface to the internals and the electrodes by driving force caused by concentration difference. Certainly, such diffusion occurs through the capillary channels of the foam material, as mentioned in the manuscript “**the captured water in the liquid phase is transferred to the surfaces of the electrodes via diffusion and subsequently split into hydrogen and oxygen in situ**” on page 4.

The reservoir at the anode is used to store excess liquid. When the R.H. rises, the rate of water uptake will increase that will decrease the concentration of the electrolyte and its viscosity, hence increasing the water splitting rate given the same working voltage. We discussed this point a couple of times in the last manuscript that “**our DAE module is intrinsically self-converged, compatible with a broad range of air humidity and current density**” on page 8. However, we learnt there is still a chance that the electrolyte will become swollen under excessive humidity or low current. In this case, to avoid overflow of the electrolyte, we designed a “reservoir” between the bottom endplate and the porous foam to buffer the volume of the electrolyte. In the meantime, we agree with the comment by the reviewer that this reservoir acts as an air barrier because it also forms a part of the liquid seal of the anode.

Based on your comments, we have added the following sentence in the revised manuscript on page 4 to clarify the function of the solution reservoir.

“**The reservoir between the endplate and the porous foam (Fig.S5b) works as an air barrier and a buffer for the volume of the ionic solution at excessive fluctuation of the air humidity. This reservoir avoids the overflow of the electrolyte from the DAE module or the dry-up of the wetted foam.**”

c) In 'Supplementary Note 1' (S.I.) a height of 30m is used in eq12 which results in only a slight decrease in R.H. The 30m feels handpicked to reach the desired results. The analysis should be done more carefully.

RE: Thanks for the remark. The height of the atmosphere that contributes to the water uptake of our direct air electrolyser is an interesting question. In principle, the whole convection layer of the atmosphere, i.e. troposphere, should play a long term role, which has a thickness of around 10 km. However, considering the surface layer of the atmosphere immediately above the earth surface typically having a turbulence in the scale of tens of meters, subject to the solar radiation and the landscapes, our pick of 30m is not groundless. Following your advice, we recalculated the change of R.H. by using an atmospheric surface layer of 20m according to the literature¹. In comparison, we also include the results using a surface layer of 10 m. In the case of 30m, 20m, and 10m, the R.H. was reduced from 25% to 24.995%, 24.992%, and 24.98%, respectively. Clearly, the large scale deployment of our DAE modules in arid areas will have negligible impact on the air environment of the region.

d) Spelling mistake in Fig. 1a (Solar poential)

RE: Thanks for the comment. We have corrected the typo in the figure as per below:

Revisions of the text in **Figure 2** were highlighted on page 2 of the revised manuscript.

Fig.1 Superimposed atlas of a) water risk and solar energy potential; b) water risk and wind energy potential (excluding coast areas). Separate maps are available in Fig.S1-S3.

e) Sulfuric acid is hygroscopic but I'm not sure if it can be classified as deliquescent

RE: Thanks for the comment. We agree with your suggestion. We have updated the text to include hygroscopic as a more appropriate classification:

“~~Deliquescent~~ **Hygroscopic** substances characterized with a strong affinity with water tend to extract moisture from the atmosphere at exposure, absorbing sufficient water to form an aqueous solution which is hygroscopic in nature.

In this study, we tested several ~~deliquescent~~ **hygroscopic** materials, including CH_3COOK , KOH , and H_2SO_4 , representing a salt, a base, and an acid, respectively.”

f) In SI: Compared with Fig.S20a and Fig.S20b, we can confirm that the gas production at the cathode is high purity O_2 (>99%). This should be the anode.

RE: Thanks for the comment. We have corrected the typo in the revised manuscript as below:

“Compared with **Fig.S21a** and **Fig.S21b**, we can confirm that the gas production at the **eathode anode** is high purity O₂(>99%).”

I added my comments to the summarized report of referee #1 below.

The authors claim that their work is unique because a) it can be powered by any renewable energy source; b) it is the only one that can use direct air feed to cathode and anode; c) it is the only one that can produce high purity hydrogen; d) it is the only one that works in dry environment and e) it is the only one having a STH efficiency above 20%.

Point a) is unconvincing, since any air electrolyzer can be operated with any source of power, e.g. the devices reported by Kumari, Heremans and Kistler.

Point a is addressed by the authors.

RE: We sincerely thank the reviewer for the positive comment and the satisfaction with our last response.

Point b) is not correct. The device reported here feeds air to the glass separator, not to the anode or cathode directly. It is also not clear why this would be a direct advantage. It could be indirectly beneficial for the purity of the hydrogen (see next point).

Point b is addressed by the authors.

RE: Again we sincerely thank the reviewer for the positive comment and the satisfaction with our last response.

Point c) is not entirely correct. While it is true that most reports in the past have used a flow of (typically) inert gas at the cathode, this served mostly analytical purposes (efficient evacuation of the gas for product analysis). Note that e.g. Xu et al (2019) utilize a dry gas stream at the cathode not supplying any water vapor. There is no fundamental reason why these reported devices couldn't operate without a continuous gas flow at the cathode, thus resulting in high purity hydrogen. Rongé et al. (2014) indeed did not use a continuous gas flow at the cathode, simply an initial purge. Kistler et al. (doi: 10.1149/2.0041905jes) performed experiments without any gas flow at the cathode. Thus, while it is true that most earlier devices utilized a gas flow at the cathode, one cannot conclude that it is impossible to obtain a high purity hydrogen product with those devices.

There seems to be some disagreement between the authors and referee#1 on this point so I will try to elucidate both standpoints to try to come to a resolution. In their response, the authors have demonstrated that it is not possible for the devices from the literature to reach high H₂ purity in the way these were operated. This was also demonstrated by the actual purity of the produced H₂. It looks to me that reviewer #1 has a problem with the wording of the statements in the main text

but not necessarily with the actual argument. To give an example from the text: 'none of the existing electrolyzers using vapor feed or photocatalytic water splitting can deliver >99% purity hydrogen at the cathode output'. I would say that we cannot know with certainty that these devices cannot reach high purities if they are operated in a different process setup. It would be safer to say that these devices have not been able to reach high purities with their current process setup.

RE: We are grateful to the resolution advised by the reviewer, and we fully agree that the wording of that particular sentence need to be revised to make the statement safer. Hence, we make the following revisions as highlighted on page 12 in the manuscript.

“As shown in Fig.4c, apart from our work, ~~none of~~ the existing electrolyzers using vapor feed or photocatalytic water splitting **have not been able to** deliver >99% purity hydrogen at the cathode output **with their current process setup.**”

Point d) is justified and is the main benefit of the device reported here. While others have reported good performance at R.H. down to 30%, this device has the potential to operate better than earlier devices at even lower humidities.

Point d is addressed by the authors.

RE: We thank the reviewer for the satisfaction with our last response.

Point e) is utterly misleading. Some of the earlier devices operated at electrolyzer efficiencies well above 50%, while the electrolyzer reported here achieves an energy efficiency of 43.8-52.8%. When combined with a conventional silicon solar cell, this would result in STH efficiency well below 20% (as the authors admit themselves in Supplementary Table 4). The number of 32% which is quoted in the main text is purely theoretical and based on a combination with an exotic P.V. cell at extremely high efficiency. The same theoretical argument could be made for any of the devices already reported in the literature. Moreover, it neglects the impact of coupling efficiency. Which brings me to the following point: one should only speak of STH efficiency if one has actually done the experiment and can show the data. The authors have made a demonstration with a conventional solar cell and should use these data to claim any STH efficiency in the text (which can only be done if they have measured the solar irradiance).

The authors gave a reasonable alternative definition for the theoretical STH. Because this STH value depends on the efficiency of the solar cells, it is not a conclusive indication that the DAE device performs better than other comparable devices. It would be better to compare the efficiencies of the electrolyzers. The electrolyzer constructed by the authors provides a reasonable efficiency but is not the best in the field. This is ok as the main selling point would be the air-based concept at low R.H. with high H₂ purity. And more importantly, I'd say it would be intuitively

normal to achieve lower performance at low water activity. A comparison between performances of other devices found in the literature (e.g. current density at a certain operating voltage, electrolyzer efficiency,...) specified at a specific R.H. can be valuable to the reader and this would provide a more fair comparison in terms of efficiency/performance.

RE: Thanks for the comment.

Firstly, we agree with the reviewer that the particular operating conditions used by the various hydrogen production devices from the literature need to be detailed for the comparison between the performance of our DAE and other devices. Corresponding to **Figure 4** in the main text, we have listed the performances and the detailed operating conditions of the devices in literature to the best of our knowledge in **Table S2** and also shown below, including for instance, relative humidity of the feed gas, type of (inert) carrier gas, heating temperature, light source/wavelength, efficiency of solar cell and etc. A new column with the title of “**PV type or solar efficiency (%)**” has been added to Table S2. These parameters are valuable to the readers to understand the novelty and advantage of our DAE module over hydrogen electrolyzers and get a fair comparison between different electrolyser devices.

Ref.	High purity hydrogen at output	Free of inert gas carrier at cathode	Use of direct air	Free of Photocatalyst	#Free of membrane	Suitable for (semi-)arid environment	Long-term Stability	H ₂ prod. rate (L m ⁻² h ⁻¹) [⊖]	STH (%)	PV type or solar efficiency (%)
Guo et al. (this work)	√ (>99%)	√	√	√	√	√	√	93.1	15.7	Triple-junction
2. Rongé et al. (2014) [†]	×	×	√	×	×	×	√	0.025	0.0068	Photoelectrochemical
3. Xu et al. (2019) [*]	×	×	√ humidified	×	×	×	×	/	NA	NA
4. Spurgeon et al. (2011) ^T	×	×	√ humidified	√	×	×	×	/	NA	NA
5. Kumari et al. (2016) [§]	× (<2%)	×	√ humidified	√	×	×	√ high R.H.	20.6	6.2	7.9
6. Zafeiropoulos et al.(2019) [□]	× (<1%)	×	√ humidified	×	×	×	√	1.9	NA	NA
7. Fornaciari et al. (2020) ^φ	×	×	×	√	×	×	×	/	NA	NA
8. Kistler et al. (2020) [‡]	× (<2%)	×	×	√	×	×	√	63.1	14	Triple-junction
9. Heremans et al. (2017)	× (<1%)	×	×	√	×	×	√	55.8	15.1	20.6
10. Heremans et al. (2019)	× (<1%)	×	×	√	×	×	√	45.7	NA	NA
11. Amano et al. (2018) ^ψ	× (<1%)	×	×	×	×	×	√	0.59	NA	NA
12. Amano et al. (2020) [⊖]	× (<2%)	×	×	×	×	×	√	7.3	NA	NA
13. Chen et al. (2020)	×	NA	NA	×	√	×	×	/	NA	NA
14. Daeneke et al. (2017)	×	NA	NA	×	√	×	√	1.6	0.44	Photocatalyst
15. Nishiyama et al. (2021) [⊖]	× (66%)	NA	NA	×	√	×	√	2.5	0.76	Photocatalyst

Supplementary Table 2 Comparison between Ref's works and ours in process.

Use of ion exchange membrane inevitably mixes the product H₂ gas with the feed gas resulting in very low H₂ purity.

⊖ Assume faradaic efficiency=100%.

† The cathode compartment was flushed with dry nitrogen and sealed.

* Air was artificially humidified to 80% relative humidity.

T Only demonstrated high R.H. (95%) humidified air.

§ Only demonstrated high R.H. (80%) humidified air, with the hydrogen faradaic efficiency=63%.

□ Air was artificially humidified to 60% R.H. at 30°C, or heat the electrolyzer to 50 °C and 70 °C for 30% R.H, and the PEC performance of the Ti/TiO₂ and W/WO₃ photoanodes was evaluated under LED-365 nm and LED-415 nm light illumination separately at 1.23 V vs RHE.

φ Electrolyzer was heated up to 80 °C for 30% R.H. humid argon was fed to both cathode and anode for reaching nearly 200 mA cm⁻² at 2V operating potential.

‡ Extra heater was used to heat the bubble humidifier to 70 °C; N₂ carrier gas was humidified to near saturation through a 70 °C bubble humidifier before feeding into the anode, and dry N₂ was used for cathode; outdoor experiments lack details of feed gas composition.

ψ Electrolyzer was operated with an applied voltage of 1.2V at 453 nm light irradiation

⊖ Electrolyzer was operated with an applied voltage of 1.2V at 365 nm UV irradiation

⊖ Liquid water was used, and hydrogen and oxygen production mixed, so a gas separation facility was needed.

Secondly, we have recalculated the solar-to-hydrogen (STH) efficiency of our DAE module using a more realistic commercially available solar cell as the coupling power supply, as per suggestion by the reviewer. A lower STH of 15.7% was determined through performance calculations, which is still among the highest compared with other devices. Revisions were highlighted on page 13 in the manuscript and in Section 16 of the supporting information.

“DAE coupled with a triple-junction solar panel can achieve a theoretical STH efficiency of 15.7% under different H₂SO₄ concentrations (Fig.S23), while coupling with the best performing solar panel using H₂SO₄ and KOH hygroscopic electrolyte can achieve a theoretical STH efficiency of 24.9% and 32%, respectively (see Table S1, S3 and S4 for more details).”

Supplementary Fig.23 The triple junction solar cell¹⁶ and DAE module performance under different H₂SO₄ concentration.

The Solar to hydrogen efficiency (STH) is calculated by following:

$$STH(\%) = \frac{1.23V \times \text{Current Density}}{\text{Solar Intensity}} \times \eta_{f,H_2} \quad (1)$$

η_{f,H_2} represents the faradaic efficiency for hydrogen evolution, which is assumed to be 100%, and 1.23 V represents the thermodynamic potential for water splitting at room temperature¹⁷. Under each R.H., the current density keeps at 12.8 mA · cm⁻², so the STH(%) under all R.H. is:

$$STH(\%) = \frac{1.23 V \times 12.8 \text{ mA} \cdot \text{cm}^{-2}}{100 \text{ mW} \cdot \text{cm}^{-2}} \times 100\% = 15.74\% \quad (2)$$

After all, we need to emphasize that “frequently, for those using vapor feed to the anode, artificial humidification has been employed to boost the humidity of the feed to above 60%^{2,3,5,6,8,9,12}, making them unsuitable for (semi-)arid environment. Also, the hydrogen production rates in the existing vapor fed electrolyzers are mostly lower than $65 \text{ L m}^{-2} \text{ h}^{-1}$, while our DAE prototype can reach $93.1 \text{ L m}^{-2} \text{ h}^{-1}$ (Fig 4C).”

The authors often refer to Faradaic efficiency. However, the G.C. results for experiments with sulfuric acid should be added to the supplementary material (they are currently only available to referees). Moreover, this represents but one measurement, while the authors claim certain numbers for F.E. during 12 days and even 8 months. Such claims are irrelevant if they are not backed by data. Fig. 3 shows the operating voltage in an experiment at (very low) constant current during 12 days. F.E. data should be added to these graphs. Similarly, such data should be made available for the 8-month experiment as well.

The authors addressed these comments partially. Referee #1 asked for the F.E. data in figure 3, can this be provided?

RE: We sincerely thank the reviewer for the positive comment and suggestions. As requested, we have included the Faradic efficiencies in the revised manuscript. Revisions were highlighted on pages 1 and 10 in the revised manuscript.

“A prototype of such H_2 generator has been successfully established and operated for 12 consecutive days with a stable performance at an average Faradaic efficiency around 95%.”

“As shown in Fig.3e, the concentration of H_2SO_4 fed to the module was 55.0 wt% initially, and it converged to 51.1 wt% over the first 120 hr. In the following 168 hr, the electrolyte concentration, the DAE module’s voltage, the mass transfer driving force for moisture absorption ($\Delta C = C_{\text{exp}}[51.1 \text{ wt}\%] - C^*[47.7 \text{ wt}\%] = 3.4 \text{ wt}\%$) and the H_2 Faradaic efficiency (around 95%) are all stabilized.”

Fig.3 e) Example recording of cell voltage (black symbol), H_2SO_4 concentration (red symbol), $C_{\text{H}_2\text{SO}_4}^* = 47.7 \text{ wt}\%$ (red dashed line), Faradaic efficiency (blue symbol) for DAE modules at constant current density of 15.0 mA cm^{-2} for 288 hr at 40 % R.H.

The authors' claims are often an act of cherry picking: STH efficiency is calculated based on a cell with KOH (and with H₂SO₄) at low current density, high R.H. and with an exotic, hypothetical P.V. device. The stability is based on experiments at very low current density (15 mA/cm²) with sulfuric acid. The high current density of 574 mA/cm² quoted in the abstract is based on an unstable system with KOH and melamine separator, at 60% R.H. Meanwhile, they also claim to operate at R.H. as low as 4%, which is demonstrated only at elevated temperature of 45 °C and reaching current densities around 30-45 mA/cm² at electrolyzer energy efficiency of 40-45%. Such claims are misleading and distracting. The merit of this work is the demonstration of a device based on sulfuric acid, noble metals and porous glass, which can continuously capture and split water from air even at low humidity. It is, however, not groundbreaking nor better performing than the multitude of devices presented until now.

Indeed, the performance of the device is not better than other related devices but the high purity hydrogen and the ability to operate at extremely low R.H. conditions is certainly an important characteristic. With the newly introduced concepts of using air as a water source, the ways of comparing data must also evolve e.g. the R.H. is likely another parameter that needs to be taken into account. However, it is necessary to address the other points mentioned in the above comment of referee #1. More reasonable numbers should be quoted for the actual well-performing and stable devices.

RE: We sincerely thank the reviewer for the positive and constructive remark. We agree with the referee that the low R.H. conditions, high H₂ production rate and high purity H₂ generation are key characteristics when comparing our DAE module with other vapor-fed electrolyzers.

The first suggestion of comparing various electrolyzers at their respective R.H. has been addressed by Table S2, in our earlier response.

For the second suggestion, we have quoted more reasonable and specific numbers when discussing the performance of our DAE module in the Discussion section on page 14 in the revised manuscript.

“Our direct air electrolysis (DAE) module can achieve exceptional performance under specific conditions, such as operational at as low as 4% relative humidity with H₂SO₄ hygroscopic electrolyte, or more than 12 days continuous H₂ generation at 40% relative humidity performing at a hydrogen Faradaic efficiency around 95% without any decay or attendance; while in the case of using KOH hygroscopic electrolyte and nickel foam electrodes, the current density can reach 574 mA cm⁻² at 4V and 60% R.H., or 177 mA cm⁻² at 3V and 15% R.H.”

Reference

- 1 Wallace, J. M. & Hobbs, P. V. *Atmospheric science: an introductory survey*. Vol. 92 (Elsevier, 2006).
- 2 Rongé, J. *et al.* Air-based photoelectrochemical cell capturing water molecules from ambient air for hydrogen production. *RSC Adv.* **4**, 29286, (2014).
- 3 Xu, K. *et al.* Hydrogen from wet air and sunlight in a tandem photoelectrochemical cell. *Int. J. Hydrog. Energy* **44**, 587-593, (2019).
- 4 Spurgeon, J. M. & Lewis, N. S. Proton exchange membrane electrolysis sustained by water vapor. *Energy Environ. Sci.* **4**, 2993, (2011).
- 5 Kumari, S., Turner White, R., Kumar, B. & Spurgeon, J. M. Solar hydrogen production from seawater vapor electrolysis. *Energy Environ. Sci.* **9**, 1725-1733, (2016).
- 6 Zafeiropoulos, G., Johnson, H., Kinge, S., van de Sanden, M. C. M. & Tsampas, M. N. Solar Hydrogen Generation from Ambient Humidity Using Functionalized Porous Photoanodes. *ACS Appl. Mater. Interfaces* **11**, 41267-41280, (2019).
- 7 Fornaciari, J. C. *et al.* The Role of Water in Vapor-fed Proton-Exchange-Membrane Electrolysis. *J. Electrochem. Soc.* **167**, 104508, (2020).
- 8 Kistler, T. A., Um, M. Y. & Agbo, P. Stable Photoelectrochemical Hydrogen Evolution for 1000 h at 14% Efficiency in a Monolithic Vapor-fed Device. *J. Electrochem. Soc.* **167**, 066502, (2020).
- 9 Heremans, G. *et al.* Vapor-fed solar hydrogen production exceeding 15% efficiency using earth abundant catalysts and anion exchange membrane. *Sustain. Energy Fuels* **1**, 2061-2065, (2017).
- 10 Heremans, G., Bosserez, T., Martens, J. A. & Rongé, J. Stability of vapor phase water electrolysis cell with anion exchange membrane. *Catal. Today* **334**, 243-248, (2019).
- 11 Amano, F., Shintani, A., Mukohara, H., Hwang, Y.-M. & Tsurui, K. Photoelectrochemical Gas–Electrolyte–Solid Phase Boundary for Hydrogen Production From Water Vapor. *Front. Chem.* **6**, (2018).
- 12 Amano, F. *et al.* Vapor-fed photoelectrolysis of water at 0.3 V using gas-diffusion photoanodes of SrTiO₃ layers. *Sustain. Energy Fuels* **4**, 1443-1453, (2020).
- 13 Chen, Y. *et al.* Water collection from air by ionic liquids for efficient visible-light-driven hydrogen evolution by metal-free conjugated polymer photocatalysts. *Renew. Energy* **147**, 594-601, (2020).
- 14 Daeneke, T. *et al.* Surface Water Dependent Properties of Sulfur-Rich Molybdenum Sulfides: Electrolyteless Gas Phase Water Splitting. *ACS Nano* **11**, 6782-6794, (2017).
- 15 Nishiyama, H. *et al.* Photocatalytic solar hydrogen production from water on a 100 m²-scale. *Nature* **598**, 304–307, (2021).
- 16 Yang, W.-x. *et al.* Investigation of room-temperature wafer bonded GaInP/GaAs/InGaAsP triple-junction solar cells. *Applied Surface Science* **389**, 673-678, (2016).
- 17 Nakamura, A. *et al.* A 24.4% solar to hydrogen energy conversion efficiency by combining concentrator photovoltaic modules and electrochemical cells. *Appl. Phys. Express* **8**, 107101, (2015).

Reviewer #3 (Remarks to the Author):

The authors have addressed all my comments in the revised manuscript.